⊜ | **Open Peer Review** | Clinical Microbiology | Research Article

# Application value of metagenomic next-generation sequencing based on protective bronchoalveolar lavage in nonresponding pneumonia

Yingchen Pang,[1,2] Junjin Qiu,[1,3] Hong Yang,[4] Junbao Zhang,[1,5] Jianming Mo,[1] Wendi Huang,[1] Chao Zeng,[1] Ping Xu[1]

**ABSTRACT** This study aims to explore the application value of metagenomic next-generation sequencing (mNGS) of protective bronchoalveolar lavage fluid in the differential diagnosis and pathogenetic identification of nonresponding pneumonia. This study analyzed patient symptoms, auxiliary examinations including pathogen detection, and treatment response to identify the reasons for the lack of response to initial treatment and the pathogenetic diagnosis of pulmonary infections. The diagnostic efficacy of pathogen culture and mNGS was statistically analyzed and compared based on the clinical diagnosis criteria. (1) The two most common reasons for the ineffectiveness of initial treatment in nonresponding pneumonia cases are that (i) the initial treatment did not cover the pathogenic bacteria in pulmonary infection cases and that (ii) non-infectious pulmonary diseases were responsible. The most common pathogens in pulmonary infection cases of nonresponding pneumonia are *Mycobacterium tuberculosis* (MTB), *Pneumocystis jirovecii*, *Aspergillus*, and *Pseudomonas aeruginosa*. (2) In pulmonary infectious cases, mNGS demonstrated a higher detection sensitivity for pathogenic bacteria than pathogen cultures. mNGS combined with protective bronchoalveolar lavage has good clinical application value in the accurate diagnosis of pathogens and identification of non-infectious diseases.

**IMPORTANCE** The combination of mNGS and the protective BAL technique demonstrates significant utility in accurately diagnosing pathogens and identifying non-infectious diseases. Misdiagnosis of non-infectious lung diseases as infectious lung diseases is a common factor contributing to the lack of response to initial treatment in nonresponding pneumonia patients. The most common pathogens in pulmonary infection cases of nonresponding pneumonia are MTB, *Pneumocystis jirovecii*, *Aspergillus*, and *Pseudomonas aeruginosa*.

**KEYWORDS** protective bronchoalveolar lavage, metagenomics next-generation sequencing, nonresponding pneumonia, non-infectious lung disease

Nonresponding pneumonia refers to pneumonia that does not respond well to initial treatment. According to statistics, 6%–15% of hospitalized patients with pneumonia do not respond to initial antibiotic treatment (1). The mortality rate of nonresponding pneumonia was as high as 49% (2–8). Previous studies have shown that the main causes of treatment failure in patients with pneumonia are empirical antibiotic therapy being less sensitive to common pathogens and not covering pathogenic microorganisms (3, 9). Therefore, etiological diagnosis is extremely important in the diagnosis and treatment of nonresponding pneumonia. At present, traditional methods for detecting pathogenic microorganisms have limitations in sensitivity and/or specificity (10). The etiological diagnosis of pneumonia remains unknown in a large proportion of cases (with some

**Peer Reviewer** Jinmin Ma, BGI, Beijing, China

Address correspondence to Ping Xu, Ping-xu@hotmail.com, or Chao Zeng, bdren1005@163.com.

Yingchen Pang, Junjin Qiu, and Hong Yang contributed equally to this article. The author order was determined based on the order of Chinese strokes.

The authors declare no conflict of interest.

See the funding table on p. 9.

studies indicating rates as high as 62%) despite extensive microbiological investigations (11–13). In addition, non-infectious diseases can increase the misdiagnosis rate of nonresponding pneumonia because of their pneumonia-like characteristics (3, 14). It is precisely because of the limitations of traditional etiological detection that clinicians attribute the negative results of etiological diagnosis to the limitations of etiological diagnosis, resulting in misdiagnosis.

With its continuous development, metagenomic next-generation sequencing (mNGS) has become a powerful tool for pathogen identification (15–18). It has been suggested that bronchoalveolar lavage fluid (BALF) provides increased sensitivity and specificity over sputum and blood (19–22). Because the respiratory tract is not sterile, the most glaring issue with BALF is that samples are prone to contamination with upper and lower respiratory tract flora (23–25). Notably, this issue is likely magnified by mNGS untargeted sequencing. In the literature, protective bronchoalveolar lavage (BAL) has been conducted using a protected transbronchoscopic balloon-tipped catheter to decrease the influence of upper airway tract colonization microorganisms (26). However, the identification of non-pathogenic (NP) bacteria in the lower respiratory tract needs clinical verification.

This study uses a prospective study approach, accompanied by thorough clinical follow-up validation, to investigate the etiology and pathogen composition of nonresponding pneumonia. The ultimate goal is to establish a theoretical foundation for future clinical diagnosis and treatment. This study also seeks to validate the diagnostic accuracy of mNGS and etiological cultures through comprehensive clinical judgment.

## MATERIALS AND METHODS

### Patients' enrollment

This study comprised a cohort of 291 adult patients diagnosed with nonresponding pneumonia who were admitted to Peking University Shenzhen Hospital between January 29, 2019, and January 30, 2022, as illustrated in Fig. 1. The inclusion criteria were as follows: first, patients were required to be at least 18 years of age. Second, patients must have exhibited infiltrating lesions on chest X-ray or computed tomography (CT) and at least one clinical symptom (e.g., cough, expiration, fever, chest pain or tightness, or dyspnea). Third, patients must meet at least one of the following conditions: (i) the clinical symptoms fail to improve after 72 h of initial antibiotic treatment, (ii) the pulmonary infiltrates persist for over 30 days or do not clearly resolve within 14 days after the initial antibiotic treatment, and (iii) the occurrence of acute respiratory failure necessitates ventilatory support and/or septic shock within hospital admission. Patients were excluded from the study if they satisfied any of the following criteria: (i) they exhibit an absolute contraindication for BAL, such as severe refractory hypoxia resulting in an inability to maintain sufficient oxygenation during the procedure, or they refuse to undergo bronchoalveolar lavage under a bronchoscope; (ii) they have a significant bleeding tendency or a disorder in coagulation function; (iii) they currently or recently experienced myocardial ischemia or have poorly controlled heart failure; and (iv) they are lost to follow-up.

### Sample collection and processing

Protected BALF was acquired for mNGS and traditional cultures (bacterial and fungal smear and culture) according to the chest CT location (27). Other pathogenic tests were performed on BALF, such as X-pert *Mycobacterium tuberculosis* (MTB) and rifampin resistance, acid-fast bacilli smear stain for myco-bacteria, galactomannan test, (1/3)-β-D-glucans, Cryptococcus antigen tests, polymerase chain reaction for the *Epstein–Barr virus* or *cytomegalovirus*, and immunoglobulin M for *Mycoplasma pneumoniae* or *Chlamydia pneumoniae* when patients were highly suspected of having an infection.

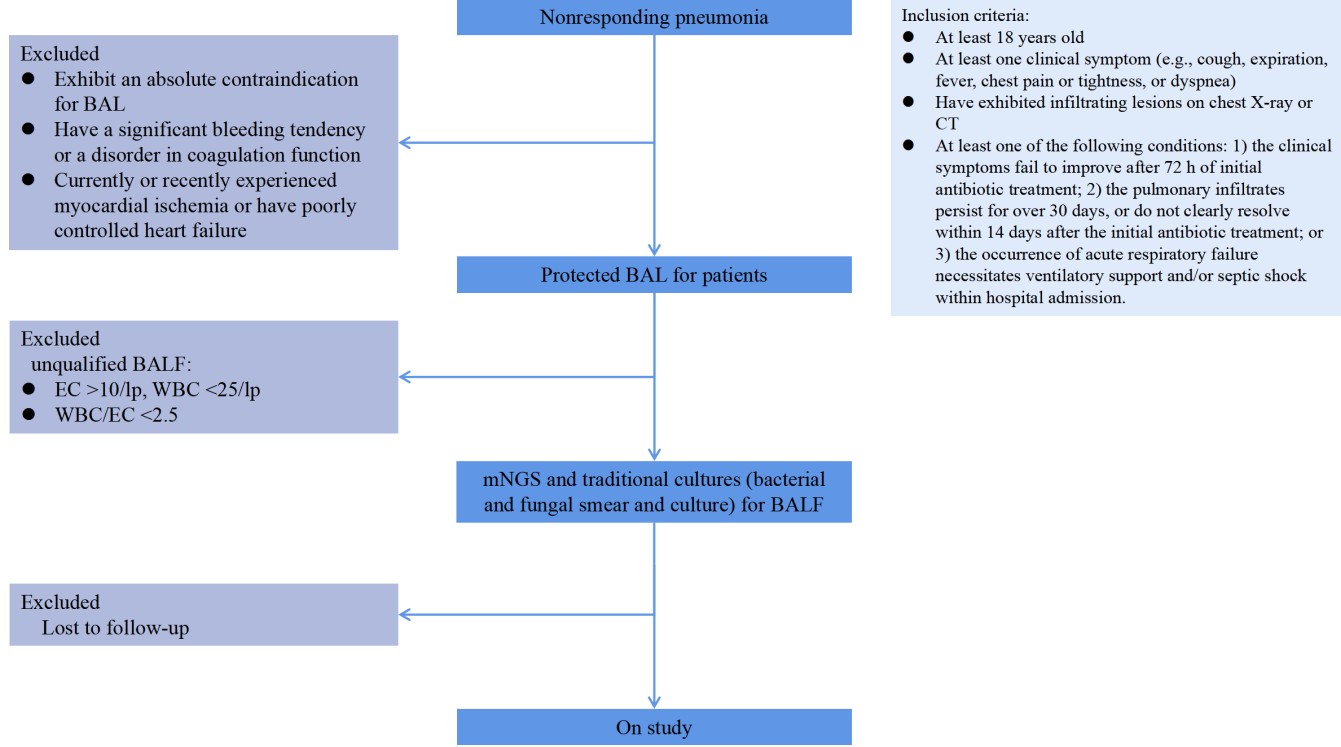

**FIG 1** Flowchart of the patient's enrollment. BAL, bronchoalveolar lavage; CT, computed tomography; WBC, white blood cell; EC, epithelial cell.

## Positive criteria for mNGS results

According to Langelier (28), bacteria (excluding mycobacteria), viruses, and parasites are deemed potential pathogens if the literature has documented their pathogenicity or if their score is at least twice as high as any other microbe of the same type identified in the patient. For fungi, mNGS identified a microbe at the species level whose coverage rate was five times higher than that of any other fungus. MTB was considered positive if at least one read was mapped to either the species or genus level. Nontuberculous mycobacteria were defined as positive when the mapping read number (genus or species level) was in the top 10 in the bacteria list.

## Follow-up and diagnosis criteria

Patients were monitored through phone or outpatient clinic appointments at specific intervals following their surgery at 1 and 2 weeks and at 1, 3, and 6 months postoperatively. The follow-up process primarily encompassed the evaluation of symptoms, signs, diagnosis, treatment progress, etiology, imaging findings, and other test results, as well as the assessment of therapeutic efficacy and prognosis. Two senior physicians conducted an exploration of the etiology and pathogen of nonresponding pneumonia, taking into consideration the patients' clinical manifestations, laboratory examinations, imaging findings, pathological results, treatment, and outcomes. In cases in which the two physicians' results were inconsistent, judgment was deferred to a third senior chief physician. Subsequently, the detection efficiency of the mNGS was evaluated based on comprehensive clinical information.

Pathogens identified through the patients' mNGS or pathogen cultures were designated as tested pathogens. If the patients' clinical symptoms showed improvement following treatment with effective antibiotics and the imaging results indicated resolution, the identified pathogen was considered a real pathogen. Conversely, the detected pathogen was considered a false positive. There are two scenarios in which a pathogen identified through mNGS or culture may be deemed a false positive. First, a

patient is ultimately diagnosed with a pulmonary infection, but the pathogen detected through mNGS or culture (which may include multiple microorganisms) is found to be non-pathogenic (hereinafter referred to as NP), such as contaminating or colonizing bacteria. Second, the patient is ultimately diagnosed with a non-infectious lung disease. Conversely, if the mNGS or etiological culture yields a negative result and the patient's diagnosis is non-infectious lung disease, it is considered a true negative; otherwise, it is classified as a false negative.

## Statistical analyses

SPSS 26.0 and Med Calc software were used for statistical analysis. Continuous variables were presented as means (standard deviations), and enumeration data were expressed as rates (%). The following parameters were used to appraise the efficacy of the diagnostic tests: sensitivity, specificity, accuracy, positive predictive value (PPV), negative predictive value (NPV), and false-negative rate (FNR). Diagnostic accuracy was compared using the areas under the receiver operating characteristic (ROC) curves (AUC). The ROC curves were also compared. $P < 0.05$ was considered statistically different.

## RESULTS

### Patients' baseline and demographic characteristics

A total of 291 patients with nonresponding pneumonia were enrolled in this study. The demographic and baseline characteristics of the patients are shown in Table 1. Following the exclusion of 17 patients (5.84%) who were lost to follow-up, 274 patients with nonresponding pneumonia successfully completed the study (Fig. 1).

### Etiology and pathogen composition of nonresponding pneumonia

The causes of a lack of response to initial treatment in this study can be broadly categorized into five groups: pathogen not encompassed by initial therapy, incorrect initial diagnosis (non-infectious lung disease), resistant pathogen, additional complications, or comorbidities (Fig. 2A). By following up on patients' clinical data and prognosis, this study revealed that out of 274 individuals with nonresponding pneumonia, 151 (55.11%) exhibited a lack of response to the initial treatment due to inadequate coverage of pathogenic bacteria but eventually achieved clinical improvement after adjusting the anti-infective treatment regimen in the later period. A total of 85 patients (31.02%) were ultimately diagnosed with non-infectious pulmonary diseases. The majority of these cases were attributed to pulmonary malignant tumors ($n = 20$, 7.30%) and organizing pneumonia ($n = 11$, 4.01%). Various other non-infectious lung diseases were identified, such as pulmonary alveolar proteinosis, allergic bronchitis, interstitial pneumonia, radiation-induced lung injury, autoimmune pneumonia, silicosis, eosinophilic pneumonia, and panbronchitis. In addition, 20 patients (7.30%) demonstrated clinical improvement solely through the extension of their anti-infection treatment duration without altering the type of antibiotics. A portion of the patients ($n = 11$, 4.01%) continued with the initial treatment due to its effectiveness in addressing the pathogenic bacteria, while another group of patients ($n = 9$, 3.28%) proceeded with the same treatment plan following negative results from pathogenic tests. Clinical improvement in these cases was achieved by extending the duration of the treatment. Six patients (2.19%) exhibited a poor basic condition, complicated by diabetes mellitus, chronic kidney disease, malignant hematological diseases, and other basic diseases. These patients did not achieve clinical improvement after initial treatment and even developed respiratory and circulatory failure. Furthermore, 12 patients failed to respond to the initial treatment due to the presence of bacterial drug resistance. No significant difference was observed in the etiology between male and female patients ($P > 0.05$). Similarly, no significant difference was found in the etiology composition among patients of different age groups ($P > 0.05$).

**TABLE 1** Baseline characteristics of patients studied[a]

| Characteristic | Value [mean (standard deviation) or n (%)] |
|---|---|
| Age | 48.91 (±15.67) |
| 18 ≥ Y < 65 | 218 (79.56%) |
| Y ≥ 65 | 56 (20.44%) |
| Sex, female | 139 (50.73%) |
| Smoking | 53 (19.34%) |
| History | |
| TB | 35 (12.77%) |
| Administration of immunosuppressives | 35 (12.77%) |
| Any comorbidity | |
| COPD | 8 (2.92%) |
| Asthma | 12 (4.38%) |
| Brochiectasis | 55 (20.07%) |
| Lung cancer | 5 (1.82%) |
| Hematologic malignancy | 19 (6.93%) |
| Autoimmunity disease | 14 (5.11%) |
| CKD | 4 (1.46%) |
| DM | 29 (10.58%) |
| HF | 1 (0.36%) |
| Symptoms | |
| Cough | 226 (82.48%) |
| Expectoration | 181 (66.06%) |
| Hemoptysis | 56 (22.44%) |
| Chest pain | 52 (18.98%) |
| Fever | 99 (36.13%) |

[a]TB, tuberculosis; COPD, chronic obstructive pulmonary disease; CKD, chronic kidney disease; DM, diabetes mellitus; HF, heart failure.

In the context of this research, 80.73% (155 out of 192) of patients with pulmonary infections were ultimately diagnosed with a clear etiology. The distribution of etiological microorganisms among a cohort of 192 patients with nonresponding pneumonia is illustrated in Fig. 2B. Bacterial, fungal, and MTB infections accounted for 32.81%, 21.35%, and 11.98% of the cases, respectively. The additional pathogens identified included *non-tuberculous mycobacteria* (NTM) (3.13%), *mycoplasma* (0.52%), *chlamydia* (0.52%), and *paragonimiasis* (0.52%). Among the bacterial infections, *Pseudomonas aeruginosa*, *anaerobes*, *actinomycetes*, *Chryseobacterium indologenes*, *Staphylococcus aureus*, and *Streptococcus pneumoniae* were the most prevalent (Fig. 2C). The most common fungal infections were *P. jiroveci*, *Aspergillus*, *Cryptococcus*, and *Candida* (Fig. 2C).

In a cohort of 151 patients whose pathogens were not covered by the initial therapeutic regimen, bacterial, fungal, and *Mycobacterium tuberculosis* (MTB) infections constituted 27.81%, 25.83%, and 15.23% of cases, respectively. Notably, *Pseudomonas aeruginosa*, actinomycetes, and anaerobic bacteria were prevalent among the bacterial infections, while *Pneumocystis jirovecii* and *Aspergillus* were frequently observed among fungal infections. Six patients experienced treatment failure due to complications, with the causative pathogens identified as Mycoplasma co-infection with *Aspergillus*, *Acinetobacter baumannii*, *Elizabethkingia*, *Micromonas*, *Przewalskii*, and *Streptococcus pneumoniae*. In contrast, among the 20 patients who showed clinical improvement following an extended course of anti-infective therapy, bacterial infections were predominant, accounting for 45% of cases. Furthermore, all 12 patients infected with drug-resistant bacteria were found to have bacterial pathogens, including *Staphylococcus aureus* (n = 1), *Pseudomonas aeruginosa* (n = 2), *Chryseobacterium indologenes* (n = 2), *Haemophilus influenzae* (n = 2), *Porphyromonas* (n = 1), *Pseudomonas putida* (n = 2), and unidentified bacteria (n = 2).

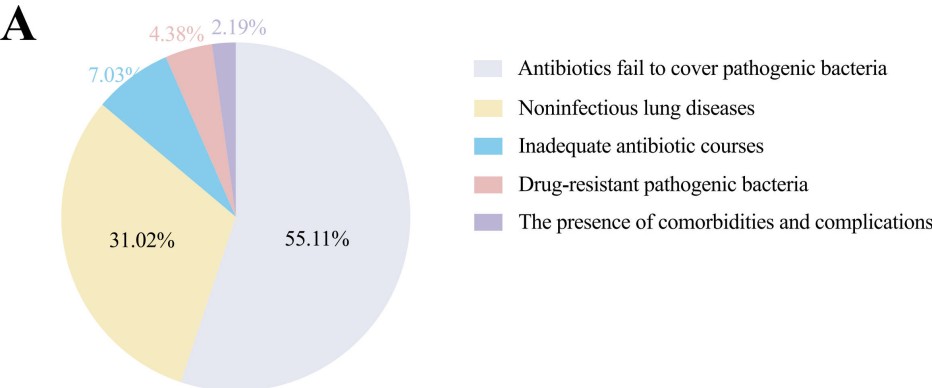

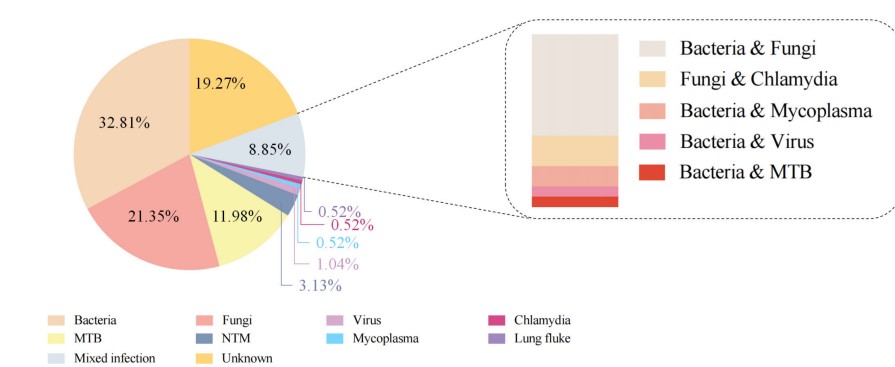

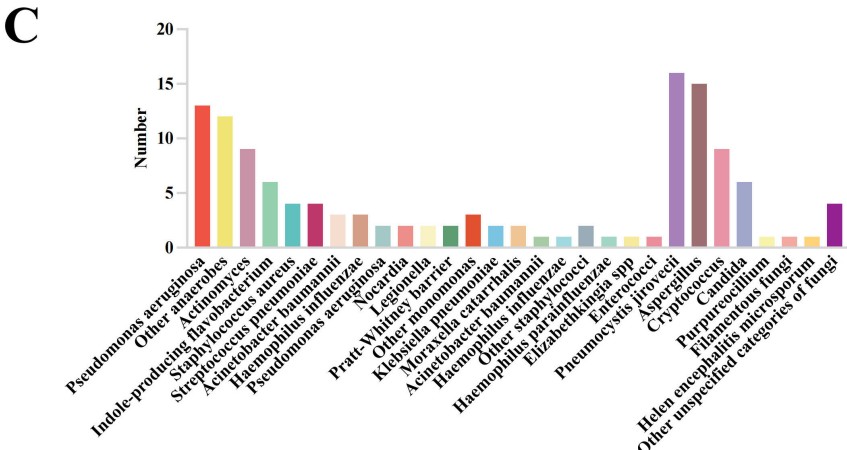

FIG 2 (A) Etiology composition of nonresponding pneumonia, (B) pathogen composition of nonresponding pneumonia; (C) identified bacterial and fungal species. NTM, nontuberculous mycobacteria; MTB, *Mycobacterium tuberculosis*.

## Comparison of the detection efficiency between mNGS and pathogenic cultures

In a cohort of 274 patients with nonresponding pneumonia, 192 were ultimately diagnosed with pulmonary infection. Tables 2 and 3 present the detection results of pathogens using the mNGS and culture methods. The study compares the diagnostic

**TABLE 2** Detection results of pathogens using mNGS[a]

|  | Infectious lung diseases | Infectious lung diseases | Total |
|---|---|---|---|
| $PTR_{mNGS}$ |  |  | 159 |
| $PM_{mNGS}$ | 136 | 0 |  |
| $NPM_{mNGS}$ | 9 | 14 |  |
| $NTR_{mNGS}$ | 47 | 68 | 115 |
| Sum | 192 | 82 | 274 |

[a]PTR, positive test result; PM, pathogenic microorganism; NPM, nonpathogenic microorganism; NTR, negative test result; mNGS, metagenomics next-generation sequencing.

efficacy of mNGS and pathogen culture in 274 patients with nonresponding pneumonia (Fig. 3A). Among patients with pulmonary infection, 70.83% (136/192) had pathogens identified by mNGS, compared with only 14.06% (27/192) by culture. The NPV of mNGS was 59.13% (68/115), higher than that of culture (32.09%), thus enabling better identification of non-infectious pulmonary diseases. The results revealed that the AUC was 0.75 for mNGS (95% confidence interval [CI], 0.68–0.81, $P < 0.001$) and 0.58 for pathogen culture (95% CI, 0.51–0.65, $P = 0.03 < 0.05$) (Fig. 3B). Furthermore, the difference in the AUC between the two methods was 0.17 (95% CI, 0.09–0.24, $P < 0.001$), indicating that mNGS exhibited significantly higher detection power than pathogen culture. The sensitivity of the combined application of mNGS and culture was 72.40%, with a NPV of 64.86%, both of which surpassed the performance of individual diagnostic methods. However, the specificity of the combined diagnostic approach was relatively low, measured at 29.27%.

## DISCUSSION

This study incorporated the clinical data and follow-up information of patients to ascertain the presence of pulmonary infectious diseases and identify specific types of pathogenic microorganisms. Ultimately, the sensitivity and specificity of mNGS were significantly higher than those of pathogen cultures. The outcome was attributed to the non-targeted approach and the high throughput of mNGS. Conversely, the findings revealed that anaerobic bacteria, MTB, fungi, and other unique pathogens constituted a significant portion of the etiological distribution in cases of nonresponding pneumonia. These pathogens are challenging to isolate through conventional pathogen culture methods, resulting in low overall detection sensitivity. Consequently, the exceptional ability of mNGS to identify these elusive pathogens underscores its superiority and confirms its diagnostic and therapeutic value in cases of nonresponding pneumonia.

In this study, both the disease diagnosis and the etiological diagnosis were definitively established. Approximately 30% of the non-responders analyzed in this study presented with non-infectious lung disease. Notably, lung cancer and organizing pneumonia emerged as the prevailing non-infectious lung diseases, both requiring prompt identification and intervention. However, the pathogenic culture method exhibited notably low sensitivity, resulting in a particularly high FNR (up to 76.04% in this investigation), rendering its negative outcomes unreliable for the identification of non-infectious diseases. This study used mNGS to determine the presence of pathogens in 159 patients with BAL samples. Among these patients, mNGS successfully detected

**TABLE 3** Detection results of pathogens using culture[a]

|  | Infectious lung diseases | Noninfectious lung diseases | Total |
|---|---|---|---|
| $PTR_{culture}$ |  |  | 59 |
| $PM_{culture}$ | 27 | 0 |  |
| $NPM_{culture}$ | 19 | 13 |  |
| $NTR_{culture}$ | 146 | 69 | 215 |
| Sum | 192 | 82 | 274 |

[a]PTR, positive test result; PM, pathogenic microorganism; NPM, nonpathogenic microorganism; NTR, negative test result.

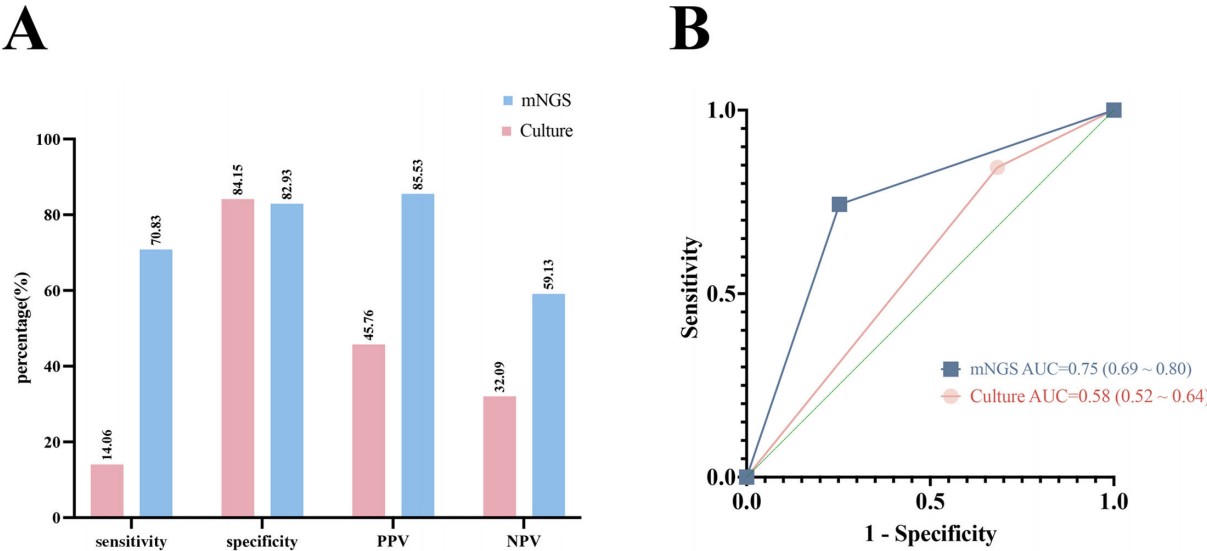

**FIG 3** (A) Diagnostic efficacy of mNGS and pathogen culture and (B) ROC curve of mNGS and culture. mNGS, metagenomics next-generation sequencing; PPV, positive predictive value; NPV, negative predictive value; AUC, calculate the areas under the curves; ROC, receiver operating characteristic.

pathogenic bacteria in 137 cases, resulting in a PPV of 85.53%. Notably, this PPV was significantly higher than that obtained from the conventional BALF pathogen culture method, which was only 45.76%. The NPV of mNGS was also found to be superior to that of the pathogen culture method, with values of 59.13% and 32.09%, respectively. The results suggest that the concurrent use of protective BAL and mNGS could present a more effective diagnostic strategy for clinicians to distinguish between infectious lung diseases and non-infectious lung diseases.

Currently, there is no existing literature on the analysis of pathogenic composition in patients with nonresponding pneumonia using mNGS. Therefore, this study attempted to combine BAL and mNGS techniques for the purpose of etiological diagnosis in patients with nonresponding pneumonia. Traditional etiological diagnosis methods and subsequent verification were also used to establish the final diagnosis for each patient. The findings indicated that the incidence of fungal and tuberculosis infections was similar to that of bacterial infections among individuals with nonresponding pneumonia. Among the 65 patients diagnosed with bacterial infection, the predominant etiological diagnoses were *P. aeruginosa* and anaerobic bacteria, followed by actinomycetes, indole-producing golden bacillus, *S. aureus*, and *S. pneumoniae*. This finding diverges from prior research, which identified *S. pneumoniae*, *Legionella*, *S. aureus*, and *P. aeruginosa* as causative agents in nonresponding pneumonia (9). The current explanation for this result lies in the ability of mNGS to identify elusive pathogens, such as anaerobic bacteria and mycobacteria, which are challenging to detect using conventional culture and molecular diagnostic techniques. In addition, all the included patients with nonresponding pneumonia had received empirical anti-infective therapy, and more common empirical anti-infective therapy, such as ceftriaxone and moxifloxacin, had covered *S. pneumoniae* or *Legionella*. Thus, in the context of diagnosing nonresponding pneumonia as an infectious pulmonary ailment, it is imperative to exercise caution in the escalation of antibiotics. Moreover, it is crucial to thoroughly explore the potential presence of *P. aeruginosa*, anaerobic bacteria, fungi, tuberculosis, and other distinctive pathogens when considering the patient's history of aspiration, contact with poultry, exposure to tuberculosis patients, and autoimmune status, as well as when conducting imaging and serological assessments.

To mitigate the contamination of bacteria colonizing the oropharynx and nasal passages, a protective BAL technique was used in this investigation, thereby enhancing the test's specificity. Nevertheless, owing to the non-sterile nature of the pulmonary

environment, bacterial colonization persists within the lower respiratory tract. The heightened sensitivity of mNGS enables the simultaneous detection of pathogenic and colonizing bacteria. Consequently, clinicians must discern between the two to facilitate a more accurate diagnosis and treatment using mNGS. Furthermore, it is worthwhile to note that the current use of mNGS does not yield insights into the drug resistance profiles of pathogenic microorganisms. The findings of this study, which identified 12 instances of drug-resistant bacterial infections, were derived solely from pathogen culture and drug sensitivity assessments. Thus, pathogen culture remains indispensable in facilitating the acquisition of information pertaining to pathogen drug resistance.

## Conclusions

The detection efficiency of protective BALF–mNGS in the etiological diagnosis was found to be superior to etiological culture, even after the administration of antibiotics.

## ACKNOWLEDGMENTS

We acknowledge support from Key Program for Clinical Research at Peking University Shenzhen Hospital (No. LCYJ2021008); Shenzhen Medical Research Fund (No. C2401006); Natural Science Foundation of Guangdong Province (No. 2023A1515012460); and Shenzhen Science and Technology Innovation Commission Foundation (No. JCYJ20240813120110015).

Y.P.: study design, data analysis and interpretation, and manuscript writing. J.Q.: data collection and analysis, and editing of manuscript tables. H.Y.: data interpretation and methodology and supervision. J.Z.: data collection and assembly and editing of manuscript figures. J.M.: patient inclusion and management and data collection. W.H.: patient inclusion and management and data collection. C.Z.: data interpretation, supervision, and manuscript revision. P.X.: study design, data interpretation, manuscript writing, and revision. All authors contributed to the article and approved the submitted version.

## AUTHOR AFFILIATIONS

[1]Respiratory Department, Peking University Shenzhen Hospital, Shenzhen, China
[2]Shenzhen Xinhua Hospital, Shenzhen, China
[3]Longgang Central Hospital, Shenzhen, China
[4]Department of Clinical Laboratory, Peking University Shenzhen Hospital, Shenzhen, China
[5]Huashan Hospital Affiliated to Fudan University, Shanghai, China

## AUTHOR ORCIDs

Chao Zeng  http://orcid.org/0000-0001-7714-3227
Ping Xu  http://orcid.org/0000-0002-8169-0394

## FUNDING

| Funder | Grant(s) | Author(s) |
| --- | --- | --- |
| Key Program for Clinical at Peking University Shenzhen Hospital | LCYJ2021008 | Ping Xu |
| Shenzhen Medical Research Fund | C2401006 | Ping Xu |
| Shenzhen Science and Technology Innovation Commmission Foundation | JCYJ20240813120110015 | Ping Xu |
| Natural Science Foundation of Guangdong Province | 2023A1515012460 | Ping Xu |

## AUTHOR CONTRIBUTIONS

Yingchen Pang, Data curation, Formal analysis, Funding acquisition, Investigation, Methodology, Supervision, Visualization, Writing – original draft, Writing – review and editing | Junjin Qiu, Data curation, Formal analysis, Supervision, Visualization, Writing – original draft | Hong Yang, Conceptualization, Data curation, Methodology, Supervision, Visualization, Writing – original draft, Writing – review and editing | Junbao Zhang, Data curation, Software, Supervision, Writing – original draft, Writing – review and editing | Jianming Mo, Data curation, Investigation, Methodology, Project administration, Resources, Supervision, Writing – review and editing | Wendi Huang, Conceptualization, Data curation, Investigation, Methodology, Project administration, Resources, Supervision, Validation, Writing – review and editing | Chao Zeng, Conceptualization, Investigation, Methodology, Project administration, Resources, Supervision, Visualization, Writing – review and editing | Ping Xu, Conceptualization, Funding acquisition, Investigation, Methodology, Project administration, Resources, Supervision, Visualization, Writing – review and editing

## DATA AVAILABILITY

The materials described in the article are available from the corresponding author upon reasonable request.

## ETHICS APPROVAL

This study was conducted in accordance with the ethical standards of Peking University Shenzhen Hospital and with the 1964 Helsinki declaration and its later amendment. The study was approved by the Ethics Committee of Peking University Shenzhen Hospital (grant number 2022108), and informed consent was obtained from all patients.

## ADDITIONAL FILES

The following material is available online.

### Open Peer Review

**PEER REVIEW HISTORY (review-history.pdf).** An accounting of the reviewer comments and feedback.

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
