## [Reviewer comments · Microbiology Spectrum]

Microbiology Spectrum

Application value of metagenomic next-generation sequencing based on protective bronchoalveolar lavage in nonresponding pneumonia.

Yingchen Pang, Junjin Qiu, Hong Yang, Junbao Zhang, Jianming Mo, Wendi Huang, Chao Zeng, and ping Xu

Corresponding Author(s): ping Xu, Peking University Shenzhen Hospital

Review Timeline:

Submission Date:	December 2, 2024
Editorial Decision:	January 23, 2025
Revision Received:	February 20, 2025
Accepted:	March 17, 2025

Editor: Sadjia Bekal

Reviewer(s): Disclosure of reviewer identity is with reference to reviewer comments included in decision letter(s). The following individuals involved in review of your submission have agreed to reveal their identity: Jinmin Ma (Reviewer #1)

Transaction Report:

DOI: <https://doi.org/10.1128/spectrum.03138-24>

Re: Spectrum03138-24 (Application value of metagenomic next-generation sequencing based on protective bronchoalveolar lavage in nonresponding pneumonia.)

Dear Dr. ping Xu:

Thank you for the privilege of reviewing your work. Below you will find my comments, instructions from the Spectrum editorial office, and the reviewer comments.

Revision Guidelines

Sincerely,
Sadjia Bekal
Editor
Microbiology Spectrum

Reviewer #1 (Comments for the Author):

This manuscript discussed the application of mNGS for nonresponding pneumonia that case high mortality rate. It is a good point for mNGS application, and the study was designed well but there were some points should be improved.

1, Table 2 and table 3 are not presented in the main manuscript, there is only table 1 marked.

2, Table 2 & 3 were not clearly for different group like "TP+mNGS", it should labeled more clearly.

- 3, The inclusion and excluded criteria in figure 1 should highlighted Intersection or concatenation.
- 4, How and why use protective bronchoalveolar lavage in the research? What is the difference between protective and non-protective bronchoalveolar lavage according to other researches?
- 5, How about the age and gender influencing the etiological? There is no information in this research, there is 20.44% patients whose age is above 65. These patients might be different.
- 6, Also, the patients were categorized into five groups: pathogen not encompassed by initial therapy, incorrect initial diagnosis (non-infectious lung disease), resistant pathogen, additional complications, or comorbidities. What is the pathogen profile in each group? What's more, how about immunosuppressives patient? Immunodeficient patients should be treated separately.
- 7, There were 31.02% patients ultimately diagnosed with non-infectious. If it can be distinguished between non-infectious and infectious according to the mNGS result ? I think it is very important to distinguish infection or not for nonresponding pneumonia cases.
- 8, It is not clear in line 191-194, it should be rewrite.
- 9, Why compare the diagnosis result between mNGS and culture? The research mentioned many other methods like PCR, IgM. What is the PPV and NPV merge all the methods ?

Reviewer #2 (Comments for the Author):

The manuscript titled "Application Value of Metagenomic Next-Generation Sequencing Based on Protective Bronchoalveolar Lavage in Nonresponding Pneumonia," authored by Chao Zeng and Ping Xu, investigates the utilization of mNGS of protective bronchoalveolar lavage for the diagnosis and pathogen identification in cases of nonresponding pneumonia. The authors demonstrate that mNGS exhibits superior detection sensitivity for pathogenic organisms. Generally speaking, the results suggest that mNGS provides more strain-specific information, facilitates the identification of novel pathogens, and holds potential for improving treatment strategies.

However, there are several issues need to be solved.

- 1 Please provide the combined sensitivity and specificity of mNGS and etiological culture for detecting causes.
- 2 To avoid confusion, it is better not to abbreviate infectious lung diseases as ILD.
- 3 Considering that the native language of all the authors is not English, they are required to have the text refined by a professional language refinement agency to improve the language quality of the full text, and a refinement certificate from a professional agency must be attached.
4. The authors should address the issues of sample size and generalizability by discussing the potential limitations and suggesting future studies with larger cohorts.

Response Letter

Dear Editors and Reviewers:

Thank you for your letter and for the reviewers' comments concerning our manuscript. Your comments are very valuable and helpful for improving our manuscript. In the following, the responses to all the comments are provided one by one.

Reviewer #1 (Comments for the Author):

This manuscript discussed the application of mNGS for nonresponding pneumonia that case high mortality rate. It is a good point for mNGS application, and the study was designed well but there were some points should be improved.

1, Table 2 and table 3 are not presented in the main manuscript, there is only table 1 marked.

Response: Thank you for your reminder. Table 2 and table 3 are located on line 236 of page 10 in the manuscript.

2, Table 2 & 3 were not clearly for different group like "TP+mNGS", it should be labeled more clearly.

Response: Thank you for your kind suggestion. We will label these different groups in page 25 and page 26: PTR, positive test result; PM, pathogenic microorganism; NPM, nonpathogenic microorganism; NTR, negative test result.

3, The inclusion and excluded criteria in figure 1 should be highlighted. Intersection or concatenation.

Response: We are grateful for your perceptive suggestion. As a result, we have made the following revisions to Figure 1.

4, How and why use protective bronchoalveolar lavage in the research?
 What is the difference between protective and non-protective bronchoalveolar lavage according to other researches?

Response: Thank you for your question. As indicated in the introduction on lines 75-89 of page 4, because the respiratory tract is not sterile, the most glaring issue with BALF is that samples are prone to contamination with upper and lower respiratory tract flora. Notably, this issue is likely magnified by mNGS untargeted sequencing. In the literature, protective bronchoalveolar lavage (BAL) has been conducted using a protected transbronchoscopic balloon-tipped catheter to decrease the influence of upper airway tract colonization microorganisms.

5, How about the age and gender influencing the etiological? There is no information in this research, there is 20.44% patients whose age is above 65. These patients might be different.

Response: Thank you for your kind suggestion. We will add these contents on lines 201-204 of page 9: No significant difference was observed in the etiology between male and female patients ($p > 0.05$). Similarly, no significant difference was found in the etiology composition among patients of different age groups ($p > 0.05$).

6, Also, the patients were categorized into five groups: pathogen not encompassed by initial therapy, incorrect initial diagnosis (non-infectious lung disease), resistant pathogen, additional complications, or comorbidities. What is the pathogen profile in each group? What's more, how about immunosuppressives patient? Immunodeficient patients should be treated separately.

Response: Thank you for your insightful advice. The investigation of the pathogenesis of pneumonia in patients with immunodeficiency constitutes the forthcoming research agenda of our research group, which will be further detailed in subsequent studies. In addition, we will add these contents on lines 216-233 of page 9 and page 10: In a cohort of 151 patients whose pathogens were not covered by the initial therapeutic regimen, bacterial, fungal, and Mycobacterium tuberculosis (MTB) infections constituted 27.81%, 25.83%, and 15.23% of cases, respectively. Notably, *Pseudomonas aeruginosa*, actinomycetes, and anaerobic bacteria were prevalent among the bacterial infections, while *Pneumocystis jirovecii* and *Aspergillus* were frequently observed among fungal infections. Six patients experienced treatment failure due to complications, with the causative pathogens identified as *Mycoplasma* co-infection with *Aspergillus*, *Acinetobacter baumannii*, *Elizabethkingia*, *Micromonas*, *Przewalskii*, and *Streptococcus pneumoniae*. In contrast, among the 20 patients who showed clinical improvement following an extended course of anti-infective therapy, bacterial infections were predominant, accounting for 45% of cases. Furthermore, all 12 patients infected with drug-resistant bacteria were found to have bacterial pathogens, including *Staphylococcus aureus* (n = 1), *Pseudomonas aeruginosa* (n = 2), *Chryseobacterium indologenes* (n = 2), *Haemophilus influenzae* (n = 2), *Porphyromonas* (n = 1), *Pseudomonas putida* (n = 2), and unidentified bacteria (n = 2).

7, There were 31.02% patients ultimately diagnosed with non-infectious.

If it can be distinguished between non-infectious and infectious according to the mNGS result? I think it is very important to distinguish infection or not for nonresponding pneumonia cases.

Response: Thank you for your kind suggestion. As indicated in the discussion on lines 274-278 on page 12 of the manuscript state: The NPV of mNGS was also found to be superior to that of the pathogen culture method, with values of 59.13% and 32.09%, respectively. The results suggest that the concurrent use of protective BAL and mNGS could present a more effective diagnostic strategy for clinicians to distinguish between infectious lung diseases and non-infectious lung diseases.

8, It is not clear in line 191-194, it should be rewrite.

Response: Thank you for your kind suggestion. We have modified the content to: A portion of the patients (n = 11, 4.01%) continued with the initial treatment due to its effectiveness in addressing the pathogenic bacteria, while another group of patients (n = 9, 3.28%) proceeded with the same treatment plan following negative results from pathogenic tests. Clinical improvement in these cases was achieved by extending the duration of the treatment.

9, Why compare the diagnosis result between mNGS and culture? The research mentioned many other methods like PCR, IgM. What is the PPV and NPV merge all the methods?

Response: Thank you for your question. Historically, the isolation and culture of bacteria and fungi from respiratory specimens have been regarded as the gold standard for diagnosing etiological agents. The other methods discussed in this study were primarily employed as adjunctive tools and were not mandated for all patients. Consequently, the positive predictive value (PPV) and negative predictive value (NPV) of these methods were not thoroughly examined. In response to recommendations from review experts, we aim to address this limitation in future research.

Reviewer #2 (Comments for the Author):

The manuscript titled "Application Value of Metagenomic Next-Generation Sequencing Based on Protective Bronchoalveolar Lavage in Nonresponding Pneumonia," authored by Chao Zeng and Ping Xu, investigates the utilization of mNGS of protective bronchoalveolar lavage for the diagnosis and pathogen identification in cases of nonresponding pneumonia. The authors demonstrate that mNGS exhibits superior detection sensitivity for pathogenic organisms. Generally speaking, the results suggest that mNGS provides more strain-specific information, facilitates the identification of novel pathogens, and holds potential for improving treatment strategies.

However, there are several issues need to be solved.

1 Please provide the combined sensitivity and specificity of mNGS and etiological culture for detecting causes.

Response: Thank you for your insightful advice. We will add these contents on lines 247-250 of page 11: The sensitivity of the combined application of mNGS and culture was 72.40%, with a NPV of 64.86%, both of which surpassed the performance of individual diagnostic methods. However, the specificity of the combined diagnostic approach was relatively low, measured at 29.27%.

2 To avoid confusion, it is better not to abbreviate infectious lung diseases as ILD.

Response: Thank you for your kind suggestion. We have modified the content to: Infectious Lung Diseases; Noninfectious Lung Diseases

3 Considering that the native language of all the authors is not English, they are required to have the text refined by a professional language refinement agency to improve the language quality of the full text, and a refinement certificate from a professional agency must be attached.

Response: Thank you for your kind suggestion. I have uploaded proof of polish to the submission system

4. The authors should address the issues of sample size and generalizability by discussing the potential limitations and suggesting future studies with larger cohorts.

Response: We appreciate your insightful advice and plan to undertake a study with a larger sample size in the future.

Re: Spectrum03138-24R1 (Application value of metagenomic next-generation sequencing based on protective bronchoalveolar lavage in nonresponding pneumonia.)

Dear Dr. ping Xu:

Your manuscript has been accepted, and I am forwarding it to the ASM production staff for publication. Your paper will first be checked to make sure all elements meet the technical requirements. ASM staff will contact you if anything needs to be revised before copyediting and production can begin. Otherwise, you will be notified when your proofs are ready to be viewed.

Sincerely,
Sadjia Bekal
Editor
Microbiology Spectrum